SciPost Physics

Submission

# Generalized eigenproblem without fermion doubling for Dirac fermions on a lattice

M. J. Pacholski,[1] G. Lemut,[1] J. Tworzydło,[2] and C. W. J. Beenakker[1]

**1** Instituut-Lorentz, Universiteit Leiden, P.O. Box 9506,
2300 RA Leiden, The Netherlands
**2** Faculty of Physics, University of Warsaw, ul. Pasteura 5,
02–093 Warszawa, Poland

March 2021

## Abstract

The spatial discretization of the single-cone Dirac Hamiltonian on the surface of a topological insulator or superconductor needs a special "staggered" grid, to avoid the appearance of a spurious second cone in the Brillouin zone. We adapt the Stacey discretization from lattice gauge theory to produce a generalized eigenvalue problem, of the form $\mathcal{H}\psi = E\mathcal{P}\psi$, with Hermitian tight-binding operators $\mathcal{H}$, $\mathcal{P}$, a locally conserved particle current, and preserved chiral and symplectic symmetries. This permits the study of the spectral statistics of Dirac fermions in each of the four symmetry classes A, AII, AIII, and D.

# 1 Introduction

Three-dimensional topological insulators are Nature's way of working around the Nielsen-Ninomiya no-go theorem [1], which forbids the existence of a single species of massless Dirac fermions on a lattice. The fermion doubling required by the theorem is present in a topological insulator slab, but the two species of Dirac fermions are spatially separated on opposite surfaces [2,3]. On each surface the two-dimensional (2D) Dirac Hamiltonian

$$H_{\mathrm{D}} = \hbar v_{\mathrm{F}} \boldsymbol{k} \cdot \boldsymbol{\sigma} = -i\hbar v_{\mathrm{F}} \left( \sigma_x \frac{\partial}{\partial x} + \sigma_y \frac{\partial}{\partial y} \right) \tag{1.1}$$

emerges as the effective low-energy Hamiltonian, with a single Dirac cone at $\boldsymbol{k} = (k_x, k_y) = 0$.

Since it is computationally expensive to work with a three-dimensional (3D) lattice, one would like to be able to discretize the 2D Dirac Hamiltonian, without introducing a second Dirac cone. We can draw inspiration from lattice gauge theory, where a variety of strategies have been developed to avoid fermion doubling [4,5]. The condensed matter context introduces its own complications, notably the lack of translational invariance and breaking of chiral symmetry by disorder and boundaries.

In Ref. 6 it was shown how the transfer matrix of the Dirac equation in a disorder potential can be discretized without fermion doubling. This allows for efficient calculation of the conductance and other transport properties in an open system [7–9]. Here we apply the same approach to the Hamiltonian of a closed system, in order to study the spectral statistics.

The Nielsen-Ninomiya theorem forbids a local discretization of the eigenvalue problem $H_{\mathrm{D}}\psi = E\psi$ without fermion doubling and without breaking the chiral symmetry relation

$$\sigma_z H_{\mathrm{D}} = -H_{\mathrm{D}} \sigma_z. \tag{1.2}$$

One way to circumvent the no-go theorem, is to abandon the locality by introducing long-range hoppings in the discretized Dirac Hamiltonian [10]. Here we follow an alternative route, following Stacey [11], which is to work with a *generalized* eigenvalue problem

$$\mathcal{H}\psi = E\mathcal{P}\psi, \tag{1.3}$$

with *local* tight-binding operators $\mathcal{H}$ and $\mathcal{P}$ on both sides of the equation. Going beyond Ref. 11, we transform the operators $\mathcal{H}$ and $\mathcal{P}$ such that they remain, respectively, Hermitian and positive definite in the absence of translational invariance. This favors a stable and efficient numerical solution, and moreover guarantees that the resulting spectrum is real, not only in the continuum limit but at any grid size.

A key feature of our approach, compared with the more familiar approaches of Wilson fermions [12] and Susskind fermions [13], is that both the chiral symmetry (1.2) is preserved and the symplectic time-reversal symmetry [14]

$$\sigma_y H_{\mathrm{D}}^* \sigma_y = H_{\mathrm{D}}. \tag{1.4}$$

This also implies the conservation of the product of the chiral and symplectic symmetries, which is a particle-hole symmetry,

$$\sigma_x H_{\mathrm{D}}^* \sigma_x = -H_{\mathrm{D}}. \tag{1.5}$$

To demonstrate the capabilities of our approach we calculate the spectral statistics of a disordered system and show how the numerics distinguishes broken versus preserved chiral or symplectic symmetry in each of the four symmetry classes of random-matrix theory [15].

The outline of the paper is as follows: In the next section we formulate the generalized eigenproblem, first following Stacey [11] for a translationally invariant system, and then including disorder. The symmetrization that produces a Hermitian $\mathcal{H}$ and positive definite $\mathcal{P}$ is introduced in Sec. 3. The locality of the discretization scheme is demonstrated by the construction of a locally conserved current in Sec. 4. By applying different types of disorder, in scalar potential, vector potential, or mass, we can access the different symmetry classes and obtain the characteristic spectral statistics for each, as we show in Sec. 5. We conclude in Sec. 6.

# 2 Construction of the generalized eigenproblem

## 2.1 Staggered discretization

If we discretize the Dirac Hamiltonian (1.1) on a lattice (lattice constant $a$), the replacement of the momentum $k$ by $a^{-1} \sin ka$ produces a second Dirac cone at the edge of the Brillouin zone ($k = \pi/a$). To place our work into context, we summarize methods to remove this spurious low-energy excitation.

If one is willing to abandon the locality of the Hamiltonian, one can eliminate the fermion doubling by a discretization of the spatial derivative that involves all lattice points, $df/dx \mapsto \sum_n (-1)^n n^{-1} f(x - na)$. The resulting dispersion remains strictly linear in the first Brillouin zone. This discretization scheme goes by the name of SLAC fermions [10] in the high-energy physics literature. It has recently been implemented in a condensed matter context [16].

An alternative line of approach preserves the locality at the expense of a symmetry breaking. The simplest way is to couple the top and bottom surfaces of the topological insulator slab [17,18]. The coupling adds a momentum dependent mass term $\mu\sigma_z(1 - \cos ka)$ which gaps out the second cone, while breaking both chiral symmetry and symplectic symmetry. This is the Wilson fermion regularization of lattice gauge theory [12,19]. The product of chiral and symplectic symmetry is preserved by Wilson fermions, which may be sufficient for some applications [20,21].

It is possible to maintain the chiral symmetry by discretizing the Dirac Hamiltonian on a pair of staggered grids. Much of the lattice gauge theory literature is based on the Susskind discretization [13], which applies a different grid to each of the two components of the spinor wave function $\psi$. On a 2D lattice it reduces the number of Dirac cones in the Brillouin zone from 4 to 2. Chiral symmetry is preserved, but symplectic symmetry is broken by the Susskind discretization (see App. A).

Hammer, Pötz, and Arnold [22,23] have developed an ingenious single-cone discretization method for the *time-dependent* Dirac equation. As in the Susskind discretization, different grids are used for each of the spinor components, but these are staggered not only in space but also in time. While this method is well suited for dynamical simulations [24,25], it is not easily adapted to energy-resolved spectral studies.

An altogether different approach, introduced by Stacey [11,26], is to evade the fermion-doubling no-go theorem by the replacement of the conventional eigenvalue problem $H_D\psi = E\psi$ by a generalized eigenproblem $U\psi = E\Phi\psi$. There is now no obstruction to having a local $U$ and $\Phi$ and also preserving chiral and symplectic symmetry.

The Stacey discretization of the transfer matrix was implemented in Ref. 6. In what follows we show how to apply it to the Hamiltonian, to solve the time-independent Dirac equation on a 2D lattice. In the next subsection we first summarize the results of Ref. 11 for a translationally invariant system, and then will present the modifications needed to

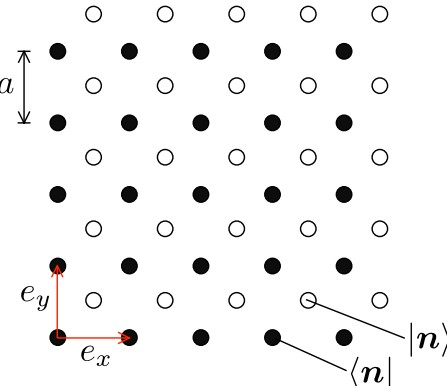

Figure 1: A pair of staggered grids (lattice constant $a$, lattice vectors $e_x, e_y$) used in the Stacey discretization of the 2D Dirac equation. The wave function and its spatial derivatives are evaluated at the open lattice points, in terms of the values on the four neighboring closed lattice points. The basis states $\langle \boldsymbol{n}|$ and $|\boldsymbol{n}\rangle$ on the two lattices are indicated.

apply the method in the presence of a disorder potential.

## 2.2 Translationally invariant system

We seek to discretize the Dirac equation $H_D \psi = E\psi$ on a 2D square lattice (lattice constant $a$). We denote the discretized wave function by $\psi_{\boldsymbol{n}}$, with $\boldsymbol{n} = (n_x, n_y) \in \mathbb{Z}^2$ labeling the lattice points at $n_x e_x + n_y e_y$. For ease of notation we will henceforth set $v_F$, $\hbar$, and $a$ to unity.

Staggered discretization *a la* Stacey means that the wave function and its spatial derivatives are evaluated on a displaced lattice with sites at the center of the unit cells of the original lattice (see Fig. 1). The discretization rules are:

$$\frac{\partial \psi}{\partial x} \mapsto \tfrac{1}{2}(\psi_{\boldsymbol{n}+e_x} + \psi_{\boldsymbol{n}+e_x+e_y} - \psi_{\boldsymbol{n}} - \psi_{\boldsymbol{n}+e_y}), \tag{2.1a}$$

$$\frac{\partial \psi}{\partial y} \mapsto \tfrac{1}{2}(\psi_{\boldsymbol{n}+e_y} + \psi_{\boldsymbol{n}+e_x+e_y} - \psi_{\boldsymbol{n}} - \psi_{\boldsymbol{n}+e_x}), \tag{2.1b}$$

$$\psi \mapsto \tfrac{1}{4}(\psi_{\boldsymbol{n}} + \psi_{\boldsymbol{n}+e_x} + \psi_{\boldsymbol{n}+e_y} + \psi_{\boldsymbol{n}+e_x+e_y}). \tag{2.1c}$$

In distinction to Susskind staggering, the same discretization applies to each spinor component.

In momentum representation, $\psi(\boldsymbol{k}) = \sum_{\boldsymbol{n}} \psi_{\boldsymbol{n}} e^{-i\boldsymbol{k}\cdot\boldsymbol{n}}$, the discretized Dirac equation reads

$$U(\boldsymbol{k})\psi(\boldsymbol{k}) = E\Phi(\boldsymbol{k})\psi(\boldsymbol{k}), \tag{2.2}$$

with the $\boldsymbol{k}$-dependent operators

$$\begin{aligned} U &= -\tfrac{1}{2}i\sigma_x(e^{ik_x} - 1)(e^{ik_y} + 1) - \tfrac{1}{2}i\sigma_y(e^{ik_x} + 1)(e^{ik_y} - 1), \\ \Phi &= \tfrac{1}{4}(e^{ik_x} + 1)(e^{ik_y} + 1). \end{aligned} \tag{2.3}$$

The dispersion relation

$$E(\boldsymbol{k}) = \pm 2\sqrt{\tan^2(k_x/2) + \tan^2(k_y/2)} \tag{2.4}$$

has a single Dirac point at $\boldsymbol{k} = 0$. The Dirac point at the edge of the Brillouin zone has been converted into a pole by the Stacey discretization.

| symmetry | symplectic | chiral | particle-hole | class |
|---|---|---|---|---|
| $V \neq 0 \neq M$ | $\times$ | $\times$ | $\times$ | A |
| $V \neq 0 = M, \boldsymbol{A}$ | $\checkmark$ | $\times$ | $\times$ | AII |
| $\boldsymbol{A} \neq 0 = V, M$ | $\times$ | $\checkmark$ | $\times$ | AIII |
| $M \neq 0 = V, \boldsymbol{A}$ | $\times$ | $\times$ | $\checkmark$ | D |

Table 1: The four symmetry classes realized by single-cone Dirac fermions [15]. The table lists the broken ($\times$) and preserved ($\checkmark$) symmetries of the Dirac Hamiltonian, in the presence of a scalar potential $V$, vector potential $\boldsymbol{A}$, and mass $M$. Class A applies if at least two of the three $V, M, \boldsymbol{A}$ are nonzero.

## 2.3 Including a disorder potential

We break translational invariance by including in the Dirac equation a spatially dependent scalar potential $V\sigma_0$, vector potential $A_x\sigma_x + A_y\sigma_z$, and mass $M\sigma_z$,

$$(-i\nabla + e\boldsymbol{A}) \cdot \boldsymbol{\sigma}\psi + (V\sigma_0 + M\sigma_z)\psi = E\psi. \tag{2.5}$$

The electron charge $e$ is set to unity in what follows. The Pauli matrices $\boldsymbol{\sigma} = (\sigma_x, \sigma_y)$ and $\sigma_z$ act on the spin degree of freedom, with $\sigma_0$ the $2 \times 2$ unit matrix.

On the surface of a topological insulator the mass term represents a perpendicular magnetization. Alternatively, we can consider a 2D topological superconductor with chiral $p$-wave pair potential, described by the Bogoliubov-de Gennes (BdG) Hamiltonian

$$H_{\text{BdG}} = \left(\frac{k^2}{2m} + V - E_{\text{F}}\right)\sigma_z + v_\Delta(\boldsymbol{k} \cdot \boldsymbol{\sigma}). \tag{2.6}$$

The Pauli matrices now act on the electron-hole degree of freedom, electrons and holes are coupled by the pair potential $\propto v_\Delta$. Since this coupling is linear in momentum $k$, the quadratic kinetic energy $k^2/2m$ can be neglected near $k = 0$. The difference $V - E_{\text{F}}$ of electrostatic potential $V$ and Fermi energy $E_{\text{F}}$ then plays the role of the mass term $M$ in Eq. (2.5).

The low-energy physics of the problem is governed by three symmetry relations, the chiral symmetry (1.2), the symplectic symmetry (1.4), and the particle-hole symmetry (1.5). Chiral symmetry is preserved by $\boldsymbol{A}$ and broken by $V$ or $M$. Symplectic symmetry is preserved by $V$ and broken by $M$ or $\boldsymbol{A}$. If at least two of the three potentials $V, M, \boldsymbol{A}$ are nonzero all symmetries of the Dirac Hamiltonian are broken. Finally, if $V = 0$, $\boldsymbol{A} = 0$ while $M \neq 0$ the particle-hole symmetry (1.5) remains. Table 1 summarizes the symmetry classification [15].

The inclusion of the vector potential requires a separate consideration, in order to preserve gauge invariance. We delay that to Sec. 4, at first we only include $V$ and $M$.

To incorporate the spatially dependent terms in the discretization scheme we write the operators $U$ and $\Phi$ in the position basis. In view of the identity

$$e^{ik_\alpha} = \sum_{\boldsymbol{n}} |\boldsymbol{n}\rangle\langle\boldsymbol{n}|e^{ik_\alpha} = \sum_{\boldsymbol{n}} |\boldsymbol{n}\rangle\langle\boldsymbol{n} + e_\alpha|, \tag{2.7}$$

we have

$$U = -\tfrac{1}{2}i\sigma_x\Omega_{+-} - \tfrac{1}{2}i\sigma_y\Omega_{-+}, \quad \Phi = \tfrac{1}{4}\Omega_{++}, \tag{2.8}$$

$$\Omega_{ss'} = \sum_{\boldsymbol{n}} \left(ss'|\boldsymbol{n}\rangle\langle\boldsymbol{n}| + s|\boldsymbol{n}\rangle\langle\boldsymbol{n} + e_x| + s'|\boldsymbol{n}\rangle\langle\boldsymbol{n} + e_y| + |\boldsymbol{n}\rangle\langle\boldsymbol{n} + e_x + e_y|\right). \tag{2.9}$$

For later use we also define the factorization $\Phi = \Phi_x \Phi_y$, with commuting operators $\Phi_x, \Phi_y$ given by

$$\Phi_\alpha = \tfrac{1}{2}(e^{ik_\alpha} + 1) = \tfrac{1}{2}\sum_{\boldsymbol{n}}\left(|\boldsymbol{n}\rangle\langle\boldsymbol{n}| + |\boldsymbol{n}\rangle\langle\boldsymbol{n} + \boldsymbol{e}_\alpha|\right). \tag{2.10}$$

In these equations the ket states $|\boldsymbol{n}\rangle$ refer to sites on the displaced lattice (open lattice points in Fig. 1), while the bra states $\langle\boldsymbol{n}|$ refer to sites on the original lattice (closed lattice points). The inner product is defined such that the two sets of eigenstates of position are orthonormal, $\langle\boldsymbol{n}'|\boldsymbol{n}\rangle = \delta_{\boldsymbol{n},\boldsymbol{n}'}$.

We define the potential and mass operators,

$$V = \sum_{\boldsymbol{n}} V_{\boldsymbol{n}}|\boldsymbol{n}\rangle\langle\boldsymbol{n}|, \quad M = \sum_{\boldsymbol{n}} M_{\boldsymbol{n}}|\boldsymbol{n}\rangle\langle\boldsymbol{n}|, \tag{2.11}$$

where $V_{\boldsymbol{n}}$ and $M_{\boldsymbol{n}}$ denote the value at the open lattice point $\boldsymbol{n}$. With this notation we have the discretized Dirac equation

$$U\psi + (V\sigma_0 + M\sigma_z)\Phi\psi = E\Phi\psi. \tag{2.12}$$

The product $V\Phi\psi$ multiplies the value of $V$ on an open lattice point with the average of the values of $\psi$ on the four adjacent closed lattice points, and similarly for $M\Phi\psi$.

Eq. (2.12) is a generalized eigenvalue problem, with operators on both sides of the equation. Neither operator is Hermitian. This is problematic in a numerical implementation, and we will show in the next section how to resolve that difficulty.

# 3 Symmetrization of the generalized eigenproblem

We wish to rewrite Eq. (2.12) in the form $\mathcal{H}\psi = E\mathcal{P}\psi$, with Hermitian $\mathcal{H}$ and Hermitian positive definite $\mathcal{P}$. Such a symmetrization of the generalized eigenvalue problem allows for a stable and efficient numerical solution [27–29]. Moreover, it guarantees real eigenvalues $E$ and eigenvectors $\psi_E$ that satisfy the orthogonality relation $\langle\psi_E|\mathcal{P}|\psi'_E\rangle = 0$ if $E \neq E'$.

We multiply both sides of Eq. (2.12) by $\Phi^\dagger$ and note that $\Phi^\dagger U$ is a Hermitian operator. In position basis it reads

$$\Phi^\dagger U = -i\boldsymbol{D}\cdot\boldsymbol{\sigma}, \quad \boldsymbol{D} = (D_x, D_y), \tag{3.1a}$$

$$D_x = \tfrac{1}{8}\sum_{\boldsymbol{n}}\left(2|\boldsymbol{n}\rangle\langle\boldsymbol{n} + \boldsymbol{e}_x| + |\boldsymbol{n}\rangle\langle\boldsymbol{n} + \boldsymbol{e}_x + \boldsymbol{e}_y| + |\boldsymbol{n}\rangle\langle\boldsymbol{n} + \boldsymbol{e}_x - \boldsymbol{e}_y|\right) - \text{H.c}, \tag{3.1b}$$

$$D_y = \tfrac{1}{8}\sum_{\boldsymbol{n}}\left(2|\boldsymbol{n}\rangle\langle\boldsymbol{n} + \boldsymbol{e}_y| + |\boldsymbol{n}\rangle\langle\boldsymbol{n} + \boldsymbol{e}_x + \boldsymbol{e}_y| + |\boldsymbol{n}\rangle\langle\boldsymbol{n} + \boldsymbol{e}_y - \boldsymbol{e}_x|\right) - \text{H.c}. \tag{3.1c}$$

We thus arrive at the generalized eigenproblem

$$\begin{aligned}
\mathcal{H}\psi &= E\mathcal{P}\psi, \quad \mathcal{P} = \Phi^\dagger\Phi, \\
\mathcal{H} &= -i\boldsymbol{D}\cdot\boldsymbol{\sigma} + \Phi^\dagger(V\sigma_0 + M\sigma_z)\Phi,
\end{aligned} \tag{3.2}$$

In the translationally invariant case the operators $\mathcal{H}$ and $\mathcal{P}$ are given by

$$\begin{aligned}
\mathcal{H} &= \tfrac{1}{2}\sigma_x(1 + \cos k_y)\sin k_x + \tfrac{1}{2}\sigma_y(1 + \cos k_x)\sin k_y, \\
\mathcal{P} &= \tfrac{1}{4}(1 + \cos k_x)(1 + \cos k_y).
\end{aligned} \tag{3.3}$$

Both operators are Hermitian and $\mathcal{P}$ is also positive semi-definite. Moreover, $\mathcal{P}$ is positive definite if the edges of the Brillouin zone ($k_x$ or $k_y$ equal to $\pm\pi$) are excluded from the

spectrum. To ensure that, we can choose an odd number $N_x, N_y$ of lattice points with periodic boundary conditions in the $x$- and $y$-directions (or alternatively, even $N_x, N_y$ with antiperiodicity).

By way of illustration, we work out the expectation value

$$\langle\psi|\Phi^\dagger V\sigma_0\Phi|\psi\rangle = \sum_{\boldsymbol{n}} V_{\boldsymbol{n}}|\tfrac{1}{4}(\psi_{\boldsymbol{n}} + \psi_{\boldsymbol{n}+e_x} + \psi_{\boldsymbol{n}+e_y} + \psi_{\boldsymbol{n}+e_x+e_y})|^2, \qquad (3.4)$$

so the value of the potential on an open lattice point is multiplied by the norm squared of the average of the wave function amplitudes on the four adjacent closed lattice points.

Eq. (3.2) is local in the sense that the operators $\mathcal{H}$ and $\mathcal{P}$ only couple nearby lattice sites. It can be converted into a conventional eigenvalue problem $\tilde{\mathcal{H}}\tilde{\psi} = E\tilde{\psi}$ with $\tilde{\psi} = \Phi\psi$ and $\tilde{\mathcal{H}}$ a *nonlocal* effective Hamiltonian:

$$\tilde{\mathcal{H}} = (\Phi^\dagger)^{-1}\mathcal{H}\Phi^{-1} = U\Phi^{-1} + \sigma_0 V + M\sigma_z. \qquad (3.5)$$

In the translationally invariant case, the effective Hamiltonian reduces simply to

$$\tilde{\mathcal{H}} = 2\sigma_x \tan(k_x/2) + 2\sigma_y \tan(k_y/2). \qquad (3.6)$$

Both chiral symmetry and symplectic symmetry are preserved on the lattice if present in the continuum description: $\sigma_z\tilde{\mathcal{H}} = -\tilde{\mathcal{H}}\sigma_z$ when $V = 0 = M$, and $\sigma_y\tilde{\mathcal{H}}^*\sigma_y = \tilde{\mathcal{H}}$ when $M = 0$.

# 4 Locally conserved particle current

In real space the effective Hamiltonian (3.5) produces infinitely long-range hoppings, as in the SLAC fermion discretization [10, 16]. The transformation to the generalized eigenproblem (3.2) restores the locality of the hoppings. One might wonder whether there is a physical content to this mathematical statement. Yes there is, as we show in this section the Stacey discretization allows for the construction of a locally conserved particle current.

We define the particle number

$$\langle\tilde{\psi}|\tilde{\psi}\rangle = \langle\psi|\Phi^\dagger\Phi|\psi\rangle, \qquad (4.1)$$

corresponding to the density operator

$$\rho(\boldsymbol{n}) = \Phi^\dagger|\boldsymbol{n}\rangle\langle\boldsymbol{n}|\Phi. \qquad (4.2)$$

With reference to the two staggered grids in Fig. 1, the particle density on an open lattice point $\boldsymbol{n}$ is given by the norm squared of the average of the wave function on the four adjacent closed lattice points,

$$\langle\psi|\rho(\boldsymbol{n})|\psi\rangle = |\tfrac{1}{4}(\psi_{\boldsymbol{n}} + \psi_{\boldsymbol{n}+e_x} + \psi_{\boldsymbol{n}+e_y} + \psi_{\boldsymbol{n}+e_x+e_y})|^2. \qquad (4.3)$$

The current density operator is given by

$$j_\alpha(\boldsymbol{n}) = (\Phi_\alpha^\dagger)^{-1}\sigma_\alpha\rho(\boldsymbol{n})\Phi_\alpha^{-1}, \qquad (4.4)$$

or equivalently,

$$\begin{aligned} j_x(\boldsymbol{n}) &= \sigma_x \sum_{\boldsymbol{n}} \Phi_y^\dagger|\boldsymbol{n}\rangle\langle\boldsymbol{n}|\Phi_y, \\ j_y(\boldsymbol{n}) &= \sigma_y \sum_{\boldsymbol{n}} \Phi_x^\dagger|\boldsymbol{n}\rangle\langle\boldsymbol{n}|\Phi_x, \end{aligned} \qquad (4.5)$$

in terms of the operators $\Phi_x, \Phi_y$ defined in Eq. (2.10). The current density in the state $\psi$ then takes the form

$$
\begin{aligned}
\langle\psi|j_x(\boldsymbol{n})|\psi\rangle &= \tfrac{1}{4}(\psi_{\boldsymbol{n}} + \psi_{\boldsymbol{n}+e_y})^{\dagger}\sigma_x(\psi_{\boldsymbol{n}} + \psi_{\boldsymbol{n}+e_y}), \\
\langle\psi|j_y(\boldsymbol{n})|\psi\rangle &= \tfrac{1}{4}(\psi_{\boldsymbol{n}} + \psi_{\boldsymbol{n}+e_x})^{\dagger}\sigma_y(\psi_{\boldsymbol{n}} + \psi_{\boldsymbol{n}+e_x}).
\end{aligned}
\tag{4.6}
$$

The current density at an open lattice point is evaluated by averaging the wave function at the two nearby closed lattice points connected by an edge perpendicular to the current flow.

The local conservation law

$$
-\frac{\partial}{\partial t}\langle\psi|\rho(\boldsymbol{n})|\psi\rangle = \sum_{\alpha=x,y}\langle\psi|j_\alpha(\boldsymbol{n}+e_\alpha) - j_\alpha(\boldsymbol{n})|\psi\rangle
\tag{4.7}
$$

is derived in App. B.

Knowledge of the current operator allows us to introduce the vector potential operator $\boldsymbol{A} = \sum_{\boldsymbol{n}}\boldsymbol{A_n}|\boldsymbol{n}\rangle\langle\boldsymbol{n}|$ such that

$$
\lim_{\boldsymbol{A}\to 0}\frac{\partial\mathcal{H}}{\partial\boldsymbol{A_n}} = \boldsymbol{j}(\boldsymbol{n}).
\tag{4.8}
$$

This is satisfied if

$$
\mathcal{H} = -i\boldsymbol{D}\cdot\boldsymbol{\sigma} + \Phi^{\dagger}\big(V\sigma_0 + M\sigma_z\big)\Phi + \Phi_y^{\dagger}\sigma_x A_x\Phi_y + \Phi_x^{\dagger}\sigma_y A_y\Phi_x + \mathcal{O}(A^2).
\tag{4.9}
$$

In App. C we check that the Hamiltonian (4.9) is gauge invariant to first order in $A$. Higher order terms are nonlocal and we will not include them.

## 5   Spectral statistics

We have tested the validity and capability of the generalized eigenvalue problem by comparing the spectral statistics with predictions from random-matrix theory (RMT). Similar tests for different methods to place Dirac fermions on a lattice have been reported in the particle physics literature [30–32].

We have solved the generalized eigenproblem

$$
\begin{aligned}
\mathcal{H}\psi &= E\mathcal{P}\psi, \quad \mathcal{P} = \Phi^{\dagger}\Phi, \\
\mathcal{H} &= -i\boldsymbol{D}\cdot\boldsymbol{\sigma} + \Phi^{\dagger}\big(V\sigma_0 + M\sigma_z\big)\Phi + \Phi_y^{\dagger}\sigma_x A_x\Phi_y + \Phi_x^{\dagger}\sigma_y A_y\Phi_x
\end{aligned}
\tag{5.1}
$$

on a square lattice of size $N_x \times N_y$. Antiperiodic boundary conditions in the $x$- and $y$-direction account for the $\pi$ Berry phase accumulated by the spin when it makes one full rotation. The dimensions $N_x, N_y$ are even to ensure a positive definite $\Phi$ (no zero-mode in the spectrum). The spectrum was calculated for $5\cdot 10^4$ realizations of a random disorder, chosen independently on each site from a uniform distribution in the interval $(-\delta, \delta)$.

To access the four symmetry classes from Table 1 we took

- $A_x, A_y \equiv 0$ and random $V, M$ with $\delta = 15/\sqrt{2}$ for class A;

- $M, A_x, A_y \equiv 0$ and random $V$ with $\delta = 15$ for class AII;

- $V, M \equiv 0$ and random $A_x, A_y$ with $\delta = \tfrac{1}{4}\sqrt{2}$ for class AIII;

- $V, A_x, A_y \equiv 0$ and random $M$ with $\delta = 15$ for class D.

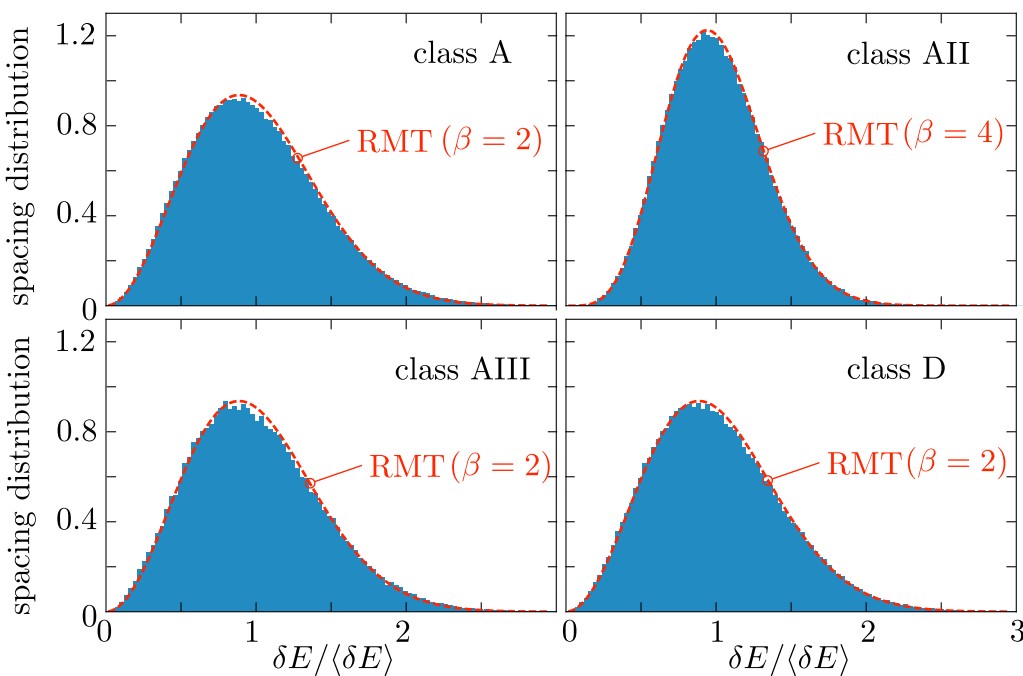

Figure 2: Histograms: Spacing distributions computed from the discretized Dirac Hamiltonian (5.1), with different types of disorder corresponding to the four symmetry classes in Table 1. The red dashed line is the prediction (5.2) from random-matrix theory in the presence of symplectic symmetry ($\beta = 4$) and in its absence ($\beta = 2$).

The relatively weak disorder in class AIII was chosen in view of the linearization in the vector potential. For that case we took $N_x = N_y = 150$, in the other symmetry classes with stronger disorder we took $N_x = N_y = 100$.

Symmetry class D is insulating for weak disorder in the mass $M \in (-\delta, \delta)$, it undergoes a metal-insulator transition at $\delta_c = 3.44$ [7]. This is the thermal metal phase of a topological superconductor [33]. The thermal metal can be reached by vortex disorder, as in the network model studied in Ref. 34, or it can be reached by electrostatic disorder in the BdG Hamiltonian (2.6), as in the tight-binding models studied in Refs. 7,35. Here we follow the latter approach, taking $\delta = 15$ much larger than $\delta_c$, so that we are deep in the metallic regime.

In Fig. 2 we show the probability distribution of the level spacing $\delta E$ in the bulk of the spectrum, far from $E = 0$, where the average spacing $\langle E \rangle$ is energy independent. We compare with the Wigner surmise from RMT [36],

$$P(s) = \begin{cases} \frac{32}{\pi^2} s^2 e^{-4s^2/\pi} & \text{in class A, AIII, D,} \\ \frac{2^{18}}{(9\pi)^3} s^4 e^{-64s^2/9\pi} & \text{in class AII,} \end{cases} \tag{5.2}$$

with $s = \delta E / \langle \delta E \rangle$. The characteristic difference between the two distributions is the decay $\propto s^\beta$ for small spacings, with $\beta = 4$ in the presence of symplectic symmetry, while $\beta = 2$ in its absence. (The case $\beta = 1$ of RMT is not realized in a spin-full system.)

In Fig. 3 we make a similar comparison for the density of states near $E = 0$. In class A and AII the ensemble averaged density of states $\rho(E)$ is flat in a broad energy range around $E = 0$. Chiral symmetry in class AIII introduces a linear dip in the density of states, while particle-hole symmetry in class D introduces a quadratic peak. The RMT

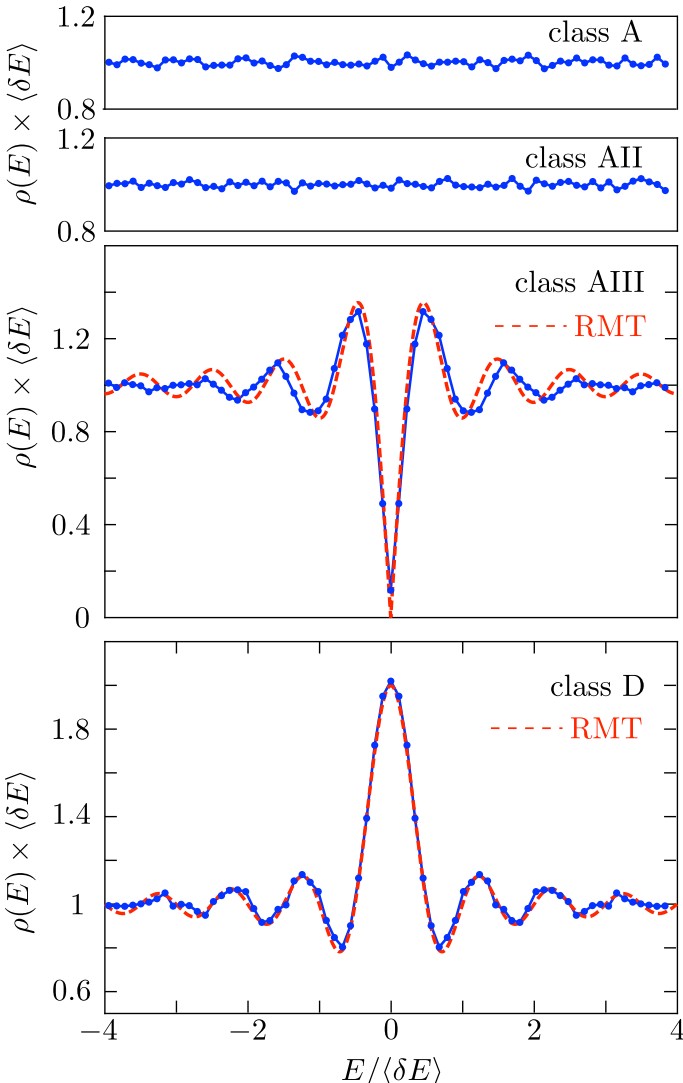

Figure 3: Density of states in the four symmetry classes, calculated numerically from the discretized Dirac Hamiltonian (blue solid lines) and compared with the RMT prediction (5.3) (red dashed lines). Chiral symmetry introduces a linear dip (class AIII), while particle-hole symmetry introduces a quadratic peak (class D).

predictions are [37, 38]

$$\rho(E) = \frac{1}{\langle \delta E \rangle} \times \begin{cases} \frac{1}{2}\pi^2 |\varepsilon| \left[ J_0^2(\pi\varepsilon) + J_1^2(\pi\varepsilon) \right] & \text{in class AIII}, \\ 1 + (2\pi\varepsilon)^{-1} \sin(2\pi\varepsilon) & \text{in class D}, \end{cases} \tag{5.3}$$

with $\varepsilon = E/\langle \delta E \rangle$. The mean level spacing $\langle \delta E \rangle$ is computed away from $E = 0$.

The good agreement between the numerical results from the disordered Dirac equation and the RMT predictions, evident in Figs. 2 and 3, is reached without any adjustable parameter. Remaining discrepancies are likely due to a dynamics that is not fully chaotic. (In particular, incipient localization can explain the shift to smaller spacings noticeable in Fig. 2.) The computer code to reproduce this data is provided [39].

# 6 Conclusion

In conclusion, we have developed and implemented a lattice fermion Hamiltonian that, unlike the familiar Wilson fermion and Susskind fermion Hamiltonians [12, 13], preserves both chiral symmetry and symplectic symmetry while avoiding fermion doubling. Our approach is a symmetrized version of Stacey's generalized eigenvalue problem [11], which allows for the construction of a locally conserved particle current. To demonstrate the universal applicability of the lattice fermion Hamiltonian we have shown how it can reproduce the characteristic spectral statistics for each of the four symmetry classes of Dirac fermions.

We mention three topics for further research. Firstly, we have only succeeded in including the vector potential in a gauge invariant way to first order, so for a flux through a unit cell that is small compared to the flux quantum. Is it possible to remove this limitation? Secondly, can we extend the approach to discretize time as well as space? And thirdly, can we incorporate boundary conditions without breaking the local current conservation?

## Acknowledgements

We have benefited from discussions with A. R. Akhmerov. This project has received funding from the Netherlands Organization for Scientific Research (NWO/OCW) and from the European Research Council (ERC) under the European Union's Horizon 2020 research and innovation programme.

## A Susskind discretization breaks symplectic symmetry

The staggered discretization of the 2D Dirac equation *a la* Susskind [13] produces a conventional eigenvalue problem, with a local Hamiltonian. There is a single Dirac cone in 1D but there are 2 Dirac cones in 2D. Chiral symmetry is preserved, but symplectic symmetry is broken. To contrast this with the symplectic-symmetry-preserving single-cone Stacey discretization used in the main text, we give a brief description of the Susskind discretization, first in 1D and then in 2D.

In 1D the staggering refers to the prescription that the derivative of the $A$ component of the spinor $\psi = (\psi_A, \psi_B)$ is calculated at $x = n + 1/2$, while the derivative of the $B$ component is calculated at $x = n - 1/2$. Hence the term $k_x \sigma_x$ in the Dirac Hamiltonian is substituted by

$$k_x \sigma_x \psi \mapsto -i \begin{pmatrix} \psi_B(n) - \psi_B(n-1) \\ \psi_A(n+1) - \psi_A(n) \end{pmatrix}$$

$$\Rightarrow H_D \mapsto -i \begin{pmatrix} 0 & 1 - e^{-\partial_x} \\ e^{\partial_x} - 1 & 0 \end{pmatrix}. \tag{A.1}$$

The exponential $e^{\partial_x}$, with $\partial_x = \partial/\partial x$, is the translation operator: $e^{\partial_x} \psi(x) = \psi(x+1)$.

In momentum representation, $\partial_x \mapsto i k_x$, the discretized Hamiltonian reads

$$H = \sigma_x \sin k_x + \sigma_y (1 - \cos k_x). \tag{A.2}$$

The corresponding dispersion relation

$$E(k_x) = \pm \sqrt{2 - 2 \cos k_x} \tag{A.3}$$

has a single Dirac cone at $k_x = 0$ in the Brillouin zone $-\pi < k_x \le \pi$.

The 2D generalization is

$$
\begin{aligned}
H_{\rm D} \mapsto & -\tfrac{1}{2}i \begin{pmatrix} 0 & (1 - e^{-\partial_x})(1 + e^{\partial_y}) \\ (e^{\partial_x} - 1)(1 + e^{-\partial_y}) & 0 \end{pmatrix} \\
& -\tfrac{1}{2}i \begin{pmatrix} 0 & -i(1 - e^{\partial_y})(1 + e^{-\partial_x}) \\ i(e^{-\partial_y} - 1)(1 + e^{\partial_x}) & 0 \end{pmatrix} \\
& = \tfrac{1}{2}\big(\sigma_x + \sigma_y\big)(\sin(k_x - k_y) - \cos k_x + \cos k_y) \\
& \quad + \tfrac{1}{2}\big(\sigma_x - \sigma_y\big)\big(\cos(k_x - k_y) + \sin k_x + \sin k_y - 1\big).
\end{aligned}
\tag{A.4}
$$

The resulting dispersion relation,

$$
E(k_x, k_y) = \pm\sqrt{2 - 2\cos k_x \cos k_y},
\tag{A.5}
$$

vanishes at $\boldsymbol{k} = (0, 0)$ and $\boldsymbol{k} = (\pi, \pi)$. (This is the dispersion studied in Ref. 40.) Without staggering there would also have been Dirac cones at $\boldsymbol{k} = (0, \pi)$ and $(\pi, 0)$, so the number of Dirac cones in the Brillouin zone has been halved by the Susskind discretization.

Chiral symmetry is preserved, $H_{\rm D}$ still anticommutes with $\sigma_z$ in its discretized form (A.4). But symplectic symmetry is broken: $\sigma_y H^* \sigma_y \ne H$ after discretization. To ensure symplectic symmetry each Pauli matrix should be multiplied by an odd function of $\boldsymbol{k}$, while Eq. (A.4) contains a mixture of odd and even functions of $\boldsymbol{k}$.

# B  Derivation of the local conservation law for the particle current

To derive Eq. (4.7) we first note the identity

$$
\frac{\partial}{\partial t} \langle \psi | O | \psi \rangle = i \langle \psi | \Phi^\dagger [\tilde{H}, \tilde{O}] \Phi | \psi \rangle,
\tag{B.1}
$$

which holds for any operator $O$, with $\tilde{O} = (\Phi^\dagger)^{-1} O \Phi^{-1}$. The nonlocal effective Hamiltonian $\tilde{H}$ is defined in Eq. (3.5).

We take for $O$ the density operator (4.2), so $\tilde{\rho}(\boldsymbol{n}) = |\boldsymbol{n}\rangle\langle\boldsymbol{n}|$. This projector commutes with the operators $V$ and $M$ in $\tilde{H}$, what remains is the commutator with $U\Phi^{-1}$:

$$
\begin{aligned}
-\frac{\partial}{\partial t} \langle \psi | \rho(\boldsymbol{n}) | \psi \rangle &= -i \langle \psi | \Phi^\dagger \big[ U\Phi^{-1}, |\boldsymbol{n}\rangle\langle\boldsymbol{n}| \big] \Phi | \psi \rangle \\
&= i \langle \psi | \Phi^\dagger |\boldsymbol{n}\rangle\langle\boldsymbol{n}| U | \psi \rangle - i \langle \psi | \Phi^\dagger U\Phi^{-1} |\boldsymbol{n}\rangle\langle\boldsymbol{n}| \Phi | \psi \rangle \\
&= i \langle \psi | \Phi^\dagger |\boldsymbol{n}\rangle\langle\boldsymbol{n}| U | \psi \rangle + \text{H.c.}
\end{aligned}
\tag{B.2}
$$

In the last equality we used that $\Phi^\dagger U = U^\dagger \Phi$.

In terms of the current operator (4.4) we have

$$
\begin{aligned}
i\Phi^\dagger |\boldsymbol{n}\rangle\langle\boldsymbol{n}| U &= \tfrac{1}{2} \sum_{\alpha=x,y} (e^{-ik_\alpha} + 1) j_\alpha(\boldsymbol{n})(e^{ik_\alpha} - 1) \\
\Rightarrow i\Phi^\dagger |\boldsymbol{n}\rangle\langle\boldsymbol{n}| U + \text{H.c} &= \sum_{\alpha=x,y} \left( e^{-ik_\alpha} j_\alpha(\boldsymbol{n}) e^{ik_\alpha} - j_\alpha(\boldsymbol{n}) \right) \\
&= \sum_{\alpha=x,y} \left( j_\alpha(\boldsymbol{n} + e_\alpha) - j_\alpha(\boldsymbol{n}) \right).
\end{aligned}
\tag{B.3}
$$

Substitution into Eq. (B.2) gives the conservation law (4.7).

## C Gauge invariant vector potential

To include the vector potential $\boldsymbol{A}(\boldsymbol{r})$ in a gauge invariant way in the discretized Dirac equation, we follow the procedure of minimal coupling: We first discretize without a vector potential, then perform a U(1) gauge transformation on the lattice, and finally replace the gradient of the phase field by the vector potential.

We define the gauge field operator

$$e^{i\theta} = \sum_{\boldsymbol{n}} e^{i\theta_{\boldsymbol{n}}} |\boldsymbol{n}\rangle\langle\boldsymbol{n}|, \tag{C.1}$$

with $\theta_{\boldsymbol{n}}$ the value of the phase $\theta(\boldsymbol{r})$ at site $\boldsymbol{n}$ on the displaced lattice (open points in Fig. 1). With this field we perform the U(1) gauge transformation

$$\begin{aligned}
\tilde{\mathcal{H}} &\mapsto e^{i\theta}\tilde{\mathcal{H}}e^{-i\theta}, \\
\Rightarrow \mathcal{H} &\mapsto \Phi^{\dagger}e^{i\theta}(\Phi^{\dagger})^{-1}\mathcal{H}\Phi^{-1}e^{-i\theta}\Phi \\
&= \Phi^{\dagger}e^{i\theta}U\Phi^{-1}e^{-i\theta}\Phi + \Phi^{\dagger}(V\sigma_0 + M\sigma_z)\Phi.
\end{aligned} \tag{C.2}$$

In the last equation we have used that $e^{i\theta}$ commutes with $V$ and $M$.

To proceed we apply the identity

$$\begin{aligned}
e^{-ik_\alpha}e^{i\theta}e^{ik_\alpha}e^{-i\theta} &= e^{i\delta_\alpha\theta}, \\
\delta_\alpha\theta &= \sum_{\boldsymbol{n}}\big(\theta(\boldsymbol{n}+e_\alpha) - \theta(\boldsymbol{n})\big)|\boldsymbol{n}\rangle\langle\boldsymbol{n}|
\end{aligned} \tag{C.3}$$

to the operator product

$$\begin{aligned}
e^{i\theta}U\Phi^{-1}e^{-i\theta} &= -2i\sum_{\alpha=x,y}\sigma_\alpha\frac{e^{i\theta}e^{ik_\alpha}e^{-i\theta}-1}{e^{i\theta}e^{ik_\alpha}e^{-i\theta}+1} \\
&= -2i\sum_{\alpha=x,y}\sigma_\alpha\frac{e^{ik_\alpha}e^{i\delta_\alpha\theta}-1}{e^{ik_\alpha}e^{i\delta_\alpha\theta}+1}.
\end{aligned} \tag{C.4}$$

The gauge transformed Hamiltonian thus takes the form

$$\mathcal{H} = \Phi^{\dagger}\left(-2i\sum_{\alpha=x,y}\sigma_\alpha\frac{e^{ik_\alpha}e^{i\delta_\alpha\theta}-1}{e^{ik_\alpha}e^{i\delta_\alpha\theta}+1} + V\sigma_0 + M\sigma_z\right)\Phi. \tag{C.5}$$

The vector potential is then introduced by the Peierls substitution

$$\theta(\boldsymbol{n}+e_\alpha) - \theta(\boldsymbol{n}) = \int_{\boldsymbol{n}}^{\boldsymbol{n}+e_\alpha}\boldsymbol{A}(\boldsymbol{r})\cdot d\boldsymbol{l}, \tag{C.6}$$

where the line integral of the vector potential is taken along a lattice bond. With this prescription the substitution can also be applied to vector potentials that do not derive from a gauge field.

The Hamiltonian (C.5) is Hermitian but nonlocal. If the phase field varies slowly on the scale of the lattice spacing, the nonlocality can be eliminated by expanding

$$e^{i\delta_\alpha\theta} \approx 1 + i\delta_\alpha\theta \equiv 1 + iA_\alpha, \quad \boldsymbol{A} = \sum_{\boldsymbol{n}}\boldsymbol{A}_{\boldsymbol{n}}|\boldsymbol{n}\rangle\langle\boldsymbol{n}|. \tag{C.7}$$

Continuing the expansion to first order in $A_\alpha$, we have

$$\frac{e^{ik_\alpha}e^{i\delta_\alpha\theta}-1}{e^{ik_\alpha}e^{i\delta_\alpha\theta}+1} = (e^{ik_\alpha}-1)(e^{ik_\alpha}+1)^{-1} + 2(e^{-ik_\alpha}+1)^{-1}iA_\alpha(e^{ik_\alpha}+1)^{-1} + \mathcal{O}(A_\alpha^2). \tag{C.8}$$

Substitution into Eq. (C.5) gives the Hamiltonian (4.9) to first order in the vector potential.

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
