# Peer review of "Generalized eigenproblem without fermion doubling for Dirac fermions on a lattice"

_SciPost Physics_

## Round 2 · Referee Report · Anonymous (Referee 1) · 2021-7-2

Report

I enjoyed refereeing the manuscript
``Generalized eigenproblem without fermion doubling
for Dirac fermions on a lattice'' by Pacholski et al.
for it is very well written, comprehensive, self-contained,
and reports interesting, new, but incremental results.

By definition, the effective minimal model at low-energies
of a three-dimensional topological insulator or superconductor
is a two-dimensional Hamiltonian describing a two-component
massless Dirac spinor bound to a surface. As such, it obeys three
symmetries called symplectic, chiral, and particle-hole.
These symmetries can be broken by single-particle or many-body effects,
in which cases four of the 10 Altland-Zirnbauer
symmetry classes of random matrix theory can be realized
in two-dimensional space.

In order to study nonpertubatively the effects of disorder or interactions
on the surface of a topological insulator,
it is necessary to rely on numerical techniques. To this end, it is desirable
to discretize the Dirac equation. However, naive discretization of the
Dirac Hamiltonian is known since the birth of lattice gauge theory in the 70's
to modify the low-energy limit by artificially increasing the rank
of the Dirac Hamiltonian (the fermion doubling problem).
It is not impossible to derive the stationary Dirac equation with a
massless Dirac Hamiltonian of rank two
from a stationary eigenvalue problem governed by a Hermitian and local lattice
Hamiltonian that shares the same three discrete symmetries as the
stationary Dirac equation.

The main result of this paper is contained in Eq. (3.3)
which describes a generalized eigenvalue problem of the form,
\begin{equation}
H\psi=E\,P\,\psi,
\end{equation}
where $H$ is a Hermitian matrix with a local representation on the lattice,
$E$ is the real-valued generalized eigenvalue,
$P=\Phi^{\dagger}\,\Phi$
is a positive semi-definite matrix with a local representation
on the lattice, and $\psi$ is the generalized eigenvector.
By construction, the continuum limit of this generalized eigenvalue problem
is the stationary Dirac equation in two-dimensional space
with a Dirac Hamiltonian of rank two that accommodates a
space-dependent scalar potential and a space-dependent mass.

Connections to the different methods used to tame the fermion-doubling problem
in lattice gauge theory is achieved by doing the nonlocal similarity
transformation
\begin{equation}
\widetilde{H}:=
(\phi^{\dagger})^{-1}\,
H\,
\Phi,
\qquad
\widetilde{\psi}:=\Phi\,\psi,
\end{equation}
in terms of which the generalized eigenvalue problem becomes the
lattice eigenvalue problem
\begin{equation}
\widetilde{H}\,\widetilde{\psi}=E\,\widetilde{\psi}.
\end{equation}
Although non-local, the Hamiltonian $\widetilde{H}$
remains invariant under all three
symplectic, chiral, and particle-hole symmetries.

It is also shown that the generalized generalized eigenvalue problem
(3.3) is compatible with a global U(1) symmetry that delivers a
local continuity equation. With this result in hand, the authors
show how to modify Eq. (3.3) so as to
accommodate a local U(1) gauge symmetry, albeit at the cost of locality
outside of the the regime of linear response theory.

This manuscript closes by studying numerically the spacing distributions
and the density of states when the generalized eigenvalue problem (3.3)
generates random eigenvalues in the symmetry classes A, AII, AIII, and D,
respectively. Both the spacing distributions and the density of states
behave as expected.

What is not done (but this could be a project of its own),
is to study the absence of localization in the symmetry class AII
for the generalized eigenvalue problem, i.e.,
the operator $P$ is not innocuous from the point of view of Anderson localization.

I strongly recommend publication of this paper as is.

(I did not understand the footnote quoted as Ref. 19. If $\mu$ is
a real-valued parameter, it is not supposed to change under reversal of time
very much like a given (as opposed to dynamical)
magnetic field in the Zeeman term is not reversed
when one performs reversal of time.)

---

## Round 2 · Referee Report · Anonymous (Referee 2) · 2021-7-13

Strengths

  1. Good numerics on large lattices.
  2. Bring attention to the Stacey discretization which is not well-know

Weaknesses

  1. References to the high-energy literature are incomplete
  2. The deviations from RMT are not explained.

Report

In this paper the authors study the spectral statics of a two-dimensional Dirac operator in a random (gauge-)potential using a discretization due to Stacey, and find agreement with predictions from random matrix universality. This is an interesting well-written paper which I can recommend for publication after the authors have addressed the following points.

  1. In the Wigner surmise we have that <s> = 1, but in Fig. 2 the histograms are systematically shifted to the left in particular for class AIII which implies that $\delta E /<\delta E>$ does not average to one, al least not for the data presented in the figure. It could be that this is due to some exceptional large spacings. For class A the deviation is small but significant, and agreement could be achieved by replacing the Wigner surmise by the exact result. Could the authors please explain the discrepancy between RMT and the numerical spacing distribution?

  2. The spectral statistics of the two-dimensional Dirac operator was studied in a closely related model by Kieburg et al. in Phys. Rev. D 90 (2014) 8, 085013, see 1405.0433. These authors also obtained the classes A, AII, AIII and D and also found that the RMT class depends on the parity of the number of lattice points in the x and y directions. The authors should cite this paper and comment on their results where relevant.

  3. Equation (5.3) for class AIII was first derived by Verbaarschot et al. in Phys. Rev. Lett. 70 (1993) 3852, see hep-th/9303012 and was first observed in (4d) lattice QCD by Gockeler et al in Phys. Rev. D 59 (1999) 094503, see hep-lat/9811018. Please add both references. Results for the Schwinger model Dirac spectra were obtained by Farchioni et al. in Nucl. Phys. B 549 (1999) 364, see hep-lat/9812018. Also include this reference.

  4. The Stacy discretization has interesting properties. Could the authors comment on whether it also works for 4d (Euclidean) gauge theory. If not, could the authors comment on the limitations of this method.

Requested changes

  1. In the Wigner surmise we have that <s> = 1, but in Fig. 2 the histograms are systematically shifted to the left in particular for class AIII which implies that $\delta E /<\delta E>$ does not average to one, al least not for the data presented in the figure. It could be that this is due to some exceptional large spacings. For class A the deviation is small but significant, and agreement could be achieved by replacing the Wigner surmise by the exact result. Could the authors please explain the discrepancy between RMT and the numerical spacing distribution?

  2. The spectral statistics of the two-dimensional Dirac operator was studied in a closely related model by Kieburg et al. in Phys. Rev. D 90 (2014) 8, 085013, see 1405.0433. These authors also obtained the classes A, AII, AIII and D and also found that the RMT class depends on the parity of the number of lattice points in the x and y directions. The authors should cite this paper and comment on their results where relevant.

  3. Equation (5.3) for class AIII was first derived by Verbaarschot et al. in Phys. Rev. Lett. 70 (1993) 3852, see hep-th/9303012 and was first observed in (4d) lattice QCD by Gockeler et al in Phys. Rev. D 59 (1999) 094503, see hep-lat/9811018. Please add both references. Results for the Schwinger model Dirac spectra were obtained by Farchioni et al. in Nucl. Phys. B 549 (1999) 364, see hep-lat/9812018. Also include this reference.

  4. The Stacy discretization has interesting properties. Could the authors comment on whether it also works for 4d (Euclidean) gauge theory. If not, could the authors comment on the limitations of this method.

---

## Round 3 · Author Response

Dear Editors, we thank the two referees for their favorable recommendation regarding the suitability of this work for SciPost Physics, and for their helpful suggestions for improvement of the manuscript.

---

## Round 3 · List of Changes

In response to Referee 1 we have removed the confusing footnote 19.

The points raised by Referee 2 have been addressed as follows:

1. The systematic shift in Fig. 2 was due to an inaccurate normalization of the numerical data, which we have now corrected. We thank the referee for spotting the shift. The numerical level spacing distribution is quite close to the RMT prediction, a remaining discrepancy (below 1%) is likely due to localization effects in the disordered quantum dot.

2. A reference to Kieburg et al. has been inserted (Ref. 32). The parity dependence mentioned by the referee is for a discretization with fermion doubling, so there is no direct connection with the single-cone discretization studied here.

3. References to Verbaarschot et al. (Ref. 37), Gockeler et al. (Ref. 30), and Farchioni et al. (Ref. 31) have been inserted.

4. In the final paragraph of the concluding section we mention three topics for future research in the context of the Stacey discretization. One of these could be the discretization of both space and time, to study dynamical properties of Dirac fermions.

We have also inserted a link to a repository that contains the computer code pertaining to this research.

You are currently on this page

Resubmission 2103.15615v3 on 15 October 2021

---

## Editorial Decision

publication_decision_taken:_accept